# `RoboMorph`: Evolving Robot Morphology using Large Language Models

Kevin Qiu[1,2], Krzysztof Ciebiera[1], Paweł Fijałkowski[1], Marek Cygan[1,3], Łukasz Kuciński[1,2,4]

[1]Univerity of Warsaw, [2]IDEAS NCBR, [3]Nomagic, [4]IMPAN

{l.qiu, k.ciebiera, p.fijalkowski, ma.cygan, l.kucinski}@uw.edu.pl

*Abstract*—We introduce `RoboMorph`, an automated approach for generating and optimizing modular robot designs using large language models (LLMs) and evolutionary algorithms. In this framework, we represent each robot design as a grammar and leverage the capabilities of LLMs to navigate the extensive robot design space, which is traditionally time-consuming and computationally demanding. By integrating automatic prompt design and a reinforcement learning based control algorithm, `RoboMorph` iteratively improves robot designs through feedback loops. Our experimental results demonstrate that `RoboMorph` can successfully generate nontrivial robots that are optimized for a single terrain while showcasing improvements in morphology over successive evolutions. Our approach demonstrates the potential of using LLMs for data-driven and modular robot design, providing a promising methodology that can be extended to other domains with similar design frameworks.

## I. INTRODUCTION

Generative methods like large language models (LLMs) have made their way into multiple domains of machine learning and everyday life, including robotics. LLMs are used to generate robot policy as code [15], provide knowledge about performing complex instructions [1], boost generalization and semantic reasoning [4, 5], design reward functions and domain radomization [18, 17], and provide generalist policies for robotic manipulation [27].

In this paper, we deal with the issue of robot design, one of the challenges of modern robotics [31]. Current engineering approaches are often time-consuming and heavily rely on a human designer to manually prototype, test and iterate over an extended design cycle. The difficulty is further increased by the vast robot design space that one needs to explore.

In this work, we propose how to address some of these challenges. We introduce `RoboMorph`, a novel approach to robot design that is data-driven, generative, and modular. `RoboMorph` is inspired by recent advancements in LLMs [22], automatic prompt design [7], and compositional design generation basing on a suitable robot grammar [34]. Additionally, we use RL-based control algorithms to compute fitness scores for each of the automatically generated robot designs.

We demonstrate experimentally that an iterative approach provides feedback to the LLM, allowing it to generate more optimal designs over time. `RoboMorph` is a proof-of-concept showcasing the viability of using LLMs to design robots, and we believe that this approach can be extended to other domains that follow a similar design framework. Our work is a step towards a future where robots are designed in a data-driven

and modular manner, enabling the rapid deployment of robots in the real world.

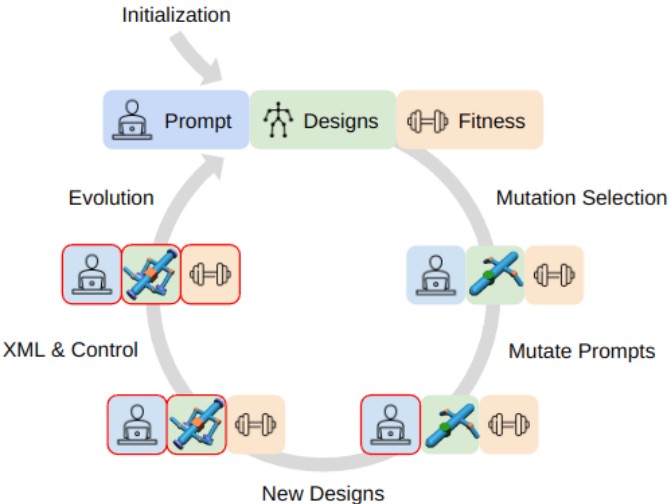

Fig. 1. Overview of `RoboMorph` framework. One iteration is visualized, where prompt (blue), robot design (green), and fitness score (orange) are mutated. The element that is changed at each step is highlighted with a red border.

Our contributions are as follows:
- We propose a proof-of-concept, `RoboMorph`, which uses LLMs, evolutionary algorithms, robot grammars and RL-based control to automate robot design for a given task.
- We provide experiments highlighting the potential of `RoboMorph` and showcasing that it is a promising approach.
- We suggest future research directions for automated robot design and `RoboMorph`.

## II. ROBOMORPH

The framework introduced in this paper, `RoboMorph`, is visualized in Figure 1. It is based on an evolutionary pipeline, which maintains a population of robot designs that was generated using an LLM (here GPT-4), together with their respective prompts and fitness scores. Each iteration starts with a binary tournament selection algorithm ([20] and Section II-B), that selects half of the population for a mutation process. Each prompt from this set is mutated in the spirit of Promptbreeder [7], see Appendix C. These mutated prompts

**Algorithm 1** Pseudo-code for `RoboMorph`

---

**Requires:** $\mathcal{D}$      min-queue of the form $\{(\texttt{fitness}_i, \texttt{design}_i, \texttt{prompt}_i)\}_{i=1}^{2K}$

        $\texttt{prompt}_{\texttt{system}}$     graph grammar rules

        $\texttt{prompt}_{\texttt{user}}$     instructions on how to design the robot

        `num_evolutions`     number of evolutions

1: **function** ROBOMORPH($\mathcal{D}, \texttt{prompt}_{\texttt{system}}, \texttt{prompt}_{\texttt{user}}, \texttt{num\_evolutions}$)
2:     **for** _ in $1, \ldots, \texttt{num\_evolutions}$ **do**
3:        Let $\pi$ be a random permutation of $1, \ldots, 2K$
4:        $\texttt{population} \leftarrow \{(\texttt{fitness}, \texttt{design}): (\texttt{fitness}, \texttt{design}, \texttt{prompt}) \in \mathcal{D}\}$
5:        **for** $i$ in $1, 3, \ldots, 2K-1$ **do**
6:           $j \leftarrow \text{argmin}_{k \in \{i, i+1\}} \texttt{fitness}_{\pi_k}$          $\triangleright$ binary tournament selection algorithm
7:           $\texttt{fitness}, \texttt{design}, \texttt{prompt} \leftarrow \texttt{fitness}_{\pi_j}, \texttt{design}_{\pi_j}, \texttt{prompt}_{\pi_j}$
8:           $\texttt{prompt}_{\texttt{new}} \leftarrow Mutate(\texttt{prompt})$
9:           $\texttt{design}_{\texttt{new}} \leftarrow LLM(\texttt{prompt}_{\texttt{system}}, \texttt{prompt}_{\texttt{new}} + \texttt{str}(\texttt{population}), \texttt{design}_{\pi_j})$
10:          $\texttt{design}_{\texttt{xml}} \leftarrow Compiler(\texttt{design}_{\texttt{new}})$
11:          **if** $\texttt{design}_{\texttt{xml}}$ is corrupted **then continue**
12:          $\texttt{fitness}_{\texttt{new}} = Evaluate(\texttt{design}_{\texttt{xml}})$
13:          Replace $\pi_j$-th entry of $\mathcal{D}$ with $(\texttt{fitness}_{\texttt{new}}, \texttt{design}_{\texttt{new}}, \texttt{prompt}_{\texttt{new}})$
14:     **return** $\mathcal{D}$

---

are then used to generate a new batch of robot designs. The designs are expressed as text which satisfies the rules of the robot grammar [34]. This is particularly useful as through recursion and branching, it allows to generate a variety of designs that are possible to construct in the real-world. Each new design is compiled to an XML file (Section II-C), fed to the MuJoCo [28] physics simulator to learn a control policy and obtain a fitness score associated to each design (Section II-D). This new batch of prompts, designs and fitness scores replace the old values in the population, and the next iteration follows. The whole pipeline is initialized with initial prompts (Section II-A and Appendix B) and resulting robot designs and fitness scores. In the following subsections, we outline each module in further detail and provide a pseudo code for `RoboMorph` in Algorithm 1.

### A. Prompt Structure and Initialization

We perform the robot design generation using GPT-4. The input to the model comprises of three parts. The first part is the *system prompt* which includes both the structural and component set of rules governed by the robot grammar [34], that ensure that the output can be physically assembled. The LLM is instructed to provide reasoning and a step-by-step process for applying the structural rules during the assembly process. The second part is the *user prompt* which instructs the LLM to design the robot. Upon selecting a design for mutation, a randomly selected mutation operator is applied to perturb the *user prompt* and generate diversity in the designs from the previous generation. Given the computational cost of methods found in evolutionary algorithms, we believe that applying the mutation on natural language is of lower fidelity than the robot design space which allows for a more efficient search.

The initial *user prompt* is a simple string: `"Design a robot"`. The last part of the prompt constitutes of the popu-lation of robots with their respective fitness scores. This allows the LLM to iterate on the existing designs and generate their improvements. For a detailed prompt listing, see Appendix B.

### B. Mutation Selection

In each iteration of `RoboMorph` we mutate half of the population chosen by a binary tournament selection algorithm [20]. This method randomly pairs all elements in the population and selects the representatives with the smaller fitness score. We choose this method over other commonly known genetic algorithms because of its ability to maintain diversity and balanced selection pressure among population candidates. These advantages are particularly well suited for our search problem as it prevents premature convergence and enables further exploration of novel designs by the LLM.

### C. Robot Design

The textual representation of a robot design, respecting the rules of the robot grammar [34], is generated by the underlying LLM. Unlike the search algorithm proposed in the original grammar [34], our method avoids the computational bottleneck of iteratively exploring multiple branches within a tree-like structure to identify the optimal design. The resulting design is compiled into an XML file compatible with the MuJoCo simulator [28]. While going directly from LLM output to XML file is possible, our initial results observed that GPT-4 was unreliable in solving this task without errors. Figure 3 presents examples of achieved designs.

### D. Control

To compute the fitness score of a design, we use the Advantage Actor-Critic (A2C) algorithm [21, 25]. In our initial experiments, A2C performed on par with more recent off-policy RL approaches [8, 10] but converged faster in total compute time due to having more environment interactions.

A2C also outperformed classical control methods such as PID [2] while not requiring design-specific fine-tuning.

## III. EXPERIMENTS

### A. Experimental Setup

We evaluate `RoboMorph` across 10 seeds, over 10 evolutions and with a population size of four. For the mutation part, we first randomly sample two mutation prompts and two thinking styles from a list provided in [7] and then select a mutation operator to apply towards the *user prompt*. We provide examples and describe the mutation operation in greater detail in Appendix C.

We assess the fitness of each design inside a custom MuJoCo environment and use the `stable_baselines3` library [23]. The states are defined by a vector representing the position and velocity of each joint, while the actions consist of the joint actuations with elements ranging within $(-1, 1)$. We define the reward function as

$$R(s, a, s') = v_x(s, a, s') + c_h \cdot \mathbb{1}_{\text{healthy}}(s'), \quad (1)$$

where $v_x(s, a, s')$ is the instantaneous velocity of the robot moving in the forward direction from state $s$ to $s'$ using action $a$ and $c_h$ is a small constant factor incentivizing the agent to remain healthy. The agent's health is determined by checking whether its $z$ coordinate value is within a predefined range.

The fitness score of a design shown in Equation 2 is defined as the average reward for the corresponding policy learned in the control module. It is estimated in a Monte-Carlo manner over $N = 15$ random rollouts

$$F(\texttt{design}) = \frac{1}{N} \sum_{i=1}^{N} \frac{1}{T_i} \sum_{j=1}^{T_i} R_{ij}, \quad (2)$$

where $T_i$ is the length of the $i$-th rollout and $R_{ij}$ is the reward obtained in the $j$-th step of the $i$-th rollout.

The architecture of the value network comprises layers of sizes [32, 64, 64, 32], while the policy network includes layers of sizes [32, 32]. We employ the AdamW optimizer [16] and ReLU activation functions across all layers in each network. The training process involves 500,000 environment interactions and uses a batch size of 512, striking a balance between computational efficiency and training stability. All other RL-related hyperparameters are maintained at their default settings as specified in the `stable_baselines3` library [23]. To ensure a fair fitness assessment of each design, these hyperparameters remain constant throughout the experiments. Additional parameters used during training are detailed in Appendix A.

### B. Results

In this section we show the results for the best robot design from the population. This element is the product of the exploitation part of our evolutionary algorithm (see Section II-B). Figure 2 presents the average fitness score of such a robot design together with the corresponding 95% confidence intervals. A positive trend indicates that each iteration of

`RoboMorph`'s loop acts as a robot design improvement operator. The confidence intervals highlight the stochastic nature of the approach, with two major sources of randomness: the RL-optimization algorithm and the evolutionary mechanism. We present the prompt associated with the best design amongst all seeds in Appendix B.

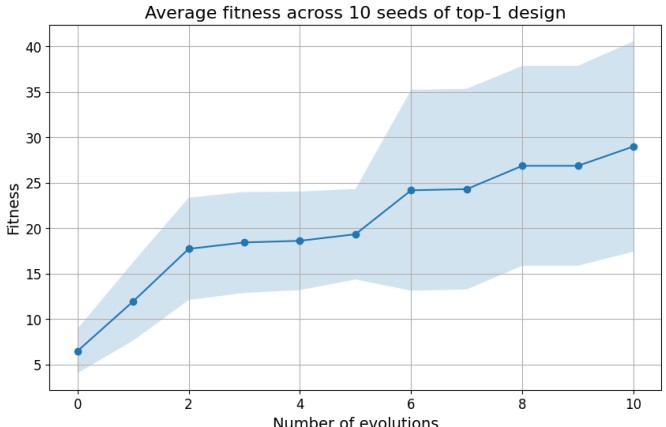

Fig. 2. Average fitness and 95% CI across 10 seeds for the best design within a population.

Figure 3 presents the top-ranked robot designs generated by `RoboMorph` for each seed. It is important to highlight that these robots are specifically tailored to our simulation environment of flat terrains and are expected to differ for other environments. A notable observation is that the limb lengths in each robot are of similar dimensions. Despite providing the LLM with a range of limb lengths to choose from, there appears to be no advantage in having limbs of varying lengths. This phenomenon is also observed in nature and could be attributed to a more even weight distribution, which enhances stability. Another common theme among all the robot designs is longer body lengths. Despite providing a wide range of body dimension values to the LLM, the robot designs tend to converge on bodies towards the longer end of the spectrum. This seems logical, as longer bodies may offer advantages for locomotion and stride length.

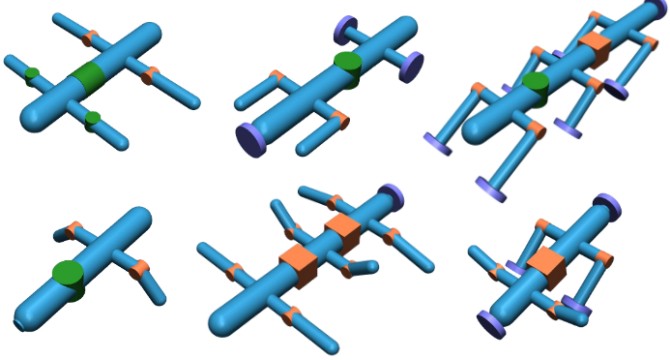

Fig. 3. Top-1 robot designs generated across 10 seeds by `RoboMorph`.

## C. Vanilla Prompting

In this section, we study the effect of removing the prompt mutation from `RoboMorph` (see Figure 1 and Section II-A). The alternation of `RoboMorph` method is dubbed *vanilla prompting*, and it generates new designs using a prompt defined as `"Design a robot"`, which remains fixed throughout all evolutions. We compare `RoboMorph` against this vanilla prompting approach and present the qualitative outcomes of the best designs generated by the two methods in Figure 4. The results could indicate that the design produced by `RoboMorph` is simpler, featuring fewer joints, which may translate to reduced complexity and lower costs in physical implementation. Moreover and in line with Occam's razor [3], the simpler design is preferred when all else is equal. The design by `RoboMorph` achieves a fitness score of 72, compared to 54 for the vanilla prompt, indicating a 33% difference. We hypothesize the higher fitness score is attributed to the effective combination of the roll joints in the front limbs and the body twist joint, suggesting potential advantages for locomotion.

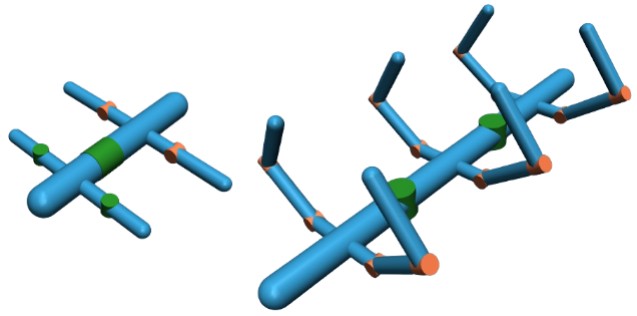

Fig. 4. Best designs generated by `RoboMorph` (left) vs. vanilla prompting (right) across 10 seeds and 10 evolutions.

## IV. RELATED WORK

**LLMs for design.** Limited work has been done in the area of LLM-based robot design, arguably considered a bottleneck in the community [31]. [26] introduces a framework using ChatGPT to guide the robot design process on both the conceptual and technical level with questions prompted by a human-in-the-loop. [19] presents an extensive study on the use of LLMs for design and manufacturing, leveraging GPT-4's ability to generate high-level structure using discrete compositions. In contrast to previous studies, our approach incorporates a fitness evaluation module within the framework which enables optimizing the design for specific objectives.

**Evolutionary algorithms**. EAs are well suited for multimodal and multi-objective problems [30] such as that found in robot design. These methods tend to be computationally expensive due to the need to evaluate robot designs in simulation over a vast design space. To address this issue, we adopt a Promptbreeder method that mutates prompts of the LLM [7] to generate diverse robot designs.

**Graph grammars.** To structure the robot design, we use robot graph grammars [24, 13, 32, 33]. In this area, we are inspired by [34], which we combine with the flexibility of LLMs to generate robot designs that are both feasible and manufacturable. A distinction between our work and that in [34] is that our approach eliminates the need to search through a design space, which we hypothesize is the computational bottleneck.

**Legged locomotion.** Designing control algorithms for legged locomotion has been a long-standing challenge in robotics that arises from the high-dimensional state space and non-linear dynamics of the system. Model predictive control (MPC) and classical control methods [6, 12] often require a detailed system dynamics model and are difficult to tune for complex systems. Consequently, we leverage a data-driven approach in reinforcement learning [11, 9, 29, 14] and specifically use the Advantage Actor-Critic (A2C) algorithm [21, 25] to learn a policy for evaluating the designs.

## V. LIMITATIONS AND FUTURE WORK

**Scale of experiments.** Since results presented in Figure 2 are promising, a future work is to scale up the experiment by increasing the number of evolution rounds, population size, and number of seeds. Additionally, further quantitative comparisons between `RoboMorph` and the vanilla prompt are to be performed.

**More tailored mutation operators.** It would be interesting to create a custom list of mutation prompts and thinking styles specific to robot design. This could address the occasional LLM's struggles to interpret the resulting mutated prompts.

**Increase of design space.** One future direction could be to increase the range of generated robot designs. This involves less stringent robot grammar rules that includes more parameters for selecting various materials, sizes, and a greater number of components to assemble robots. Another promising direction is to represent each component in the grammar as a physical hardware module, allowing for easy assembly in various configurations.

**Diverse mix of environments.** A natural further step is to ensure a broad coverage of various environmental features, such as terrain shape or texture.

**Other control algorithms.** It would be interesting to study various RL control algorithms and their impact on the generated robot design population.

**Joint generation of a robot and its policy.** It would be interesting to investigate whether the LLM can co-design both the robot morphology and the control strategy in future studies.

## VI. CONCLUSION

We present a proof-of-concept framework, `RoboMorph`, that integrates LLMs, evolutionary algorithms, RL-based control, and robot grammars for the design of modular robots. Experimental results demonstrate significant improvements in the robots' morphology over successive evolutions. Future research will focus on further exploring the intersection of the generative capabilities of LLMs and low-cost additive manufacturing techniques to design robots suitable for real-world deployment.

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

## A. Hyperparameters and Compute

We use AdamW [16] as the optimizer with $\beta$ values of (0.9, 0.999) and a weight decay of 0.01, along with ReLU activation for each layer in the policy and value networks. Notably, the value network is 4096 times larger than the policy network, significantly enhancing the model's ability to explore the action space. During training, we used a batch size of 512, 500000 environment interactions, a learning rate of 0.0007, a discount factor of 0.99, a value function coefficient of 0.5, and a maximum rollout length of 1000 steps. All experiments were executed locally, using `MPS` backend. To enable iteration of the `RoboMorph` loop, we allocated a fixed wall time of 3 minutes for learning a policy and evaluating the agent's fitness. This imposed constraints on the number of environment interactions and the sizes of both the policy and value networks.

TABLE I
ROBOMORPH ENVIRONMENT SETUP PARAMETERS

| Parameter | Value | Parameter | Value |
|---|---|---|---|
| Sim. engine | MuJoCo | Obs. dim. | $2 \times$ `num_joints` |
| Grammar | RoboGram. | Action dim. | `num_joints` |
| Seeds | 6 | Value network | [32, 64, 64, 32] |
| Evolutions | 5 | Policy network | [32, 32] |
| Population | 4 | Optimizer | AdamW |
| Mut. prompts | $2 \sim \mathcal{M}$ | Activation | ReLU |
| Mut. operators | 3 | Env. interactions | 500 000 |
| Thinking styles | $2 \sim \mathcal{T}$ | Batch size | 512 |

## B. Prompts

In this section we provide all `RoboMorph` prompts.

---

**Prompt 1: System Prompt**

```
You are a robotics engineer tasked with designing robots. My ultimate goal is to
discover as many diverse and novel designs as possible for a given task.

To construct the robot, you will represent the design in the form of a graph
consisting of nodes. I will give you the following legend where each alphabetical
letter is represented as a node in the graph:

GRAPH DEFINITION
S: start symbol
H: head part
Y: body joint
B: body part
T: tail part
U: body link
C: connector
M: mount part
E: limb end
J: limb joint
L: limb link

Given the above node definitions, you must follow the following set of structural
and component rules when building the graph of the robot.

STRUCTURAL RULES
Start node
r0) S

Body structure
r1) Replace S with H-B-T
```

```
r2) Replace T with Y-B-T

Adding appendages to the body.
r3) Replace B with U-(C-M-E)
r4) Replace B with U

Appendages
r5) Replace E with J-L-E
r6) Replace T with C-M-E
r7) Replace H with E-M-C

COMPONENT RULES
U = body=[5-40cm]
L = limb=[5-20cm]
Y = rigid, roll or twist
J = rigid, roll, knee=[0-60deg] or elbow=[0-180deg]
C = connector
M = mount
E = wheel or null
H = null
T = null
```

You are to apply rules sequentially when constructing the graph and you are to specify which structural rule you applied. Each graph begins with r0). You must replace each letter in the final structural graph with the set of component rules and only choose one parameter within each  bracket.

You must add a line break after applying each step. You should only respond in the format as described below:

RESPONSE FORMAT
STRUCTURAL RULES:
Step 1: rule) Graph
Step 2: rule) Graph
Step 3: rule) Graph
...

COMPONENT RULES:
Graph

Here's an example response after applying structural rules followed by component rules:
STRUCTURAL RULES:
Step 1: r0) S
Step 2: r1) H-B-T
Step 3: r2) H-B-Y-B-T
Step 4: 2x r3) H-U-(C-M-E)-Y-U-(C-M-E)-T
Step 5: 2x r5) H-U-(C-M-J-L-E)-Y-U-(C-M-J-L-E)-T

COMPONENT RULES
[null]-[body=10cm]-([connector]-[mount]-[knee=0deg]-[limb=10cm]-[null])-[roll]-
[body=15cm]-([connector]-[mount]-[knee=15deg]-[limb=10cm]-[null])-[null]

During the structural rule build, we apply 2x r3) denoting that r3) was applied twice in a single step so both B nodes change. Additionally, we apply 2x r5)

denoting that r5) was applied twice in a single step so both U nodes change. As shown in the example response, the structural graph is complete when all end nodes result in one of H, T or E.

During the component rule build, we convert all the nodes into individual components. The final graph must only consist of nodes denoted with components inside square brackets [ ] and not consist of any node symbols.

**Prompt 2: Initial User Prompt**

Design a robot.

**Prompt 3: Population Prompt**

Here are some examples of designs and their associated fitness values. Your task is to generate designs with the highest fitness value possible.

Example {i}:
Structural: STRUCTURAL GRAPH
Components: COMPONENT GRAPH
Fitness: FITNESS

**Prompt 4: Optimized Prompt**

You are a gifted robotics engineer, tasked with a mission of uttermost significance. Akin to ancient explorers who charted uncharted seas and discovered new lands, your voyage is to chart the boundless horizons of robotics. Your quest is to devise as many diverse and innovative designs as viable for a distinct task.

To embark on this adventure, you will have to represent the design in the form of a mysterious graph cipher, where each alphabetical character is translated as a node in this intricate web.

GRAPH DEFINITION
S: S is the rising sun that signals the beginning. It represents the start symbol.
H: H is the commander's head brimming with thoughts and convictions, it's the head piece.
Y: Y serves as the binding joint that connects the parts of the robot's chassis, it's the body joint.
B: B encapsulates the essence of the robotic form, it's the body part.
T: T plays a pivotal role as the tail part of the robot.
U: U is a conduit for connectivity, symbolizing the body link.
C: C signifies the invaluable connector.
M: M is akin to a mountaineer's base camp, the starting point of a climb, the mount part.
E: E is the edge of the world, where the possibilities end or begin, it's the limb end.
J: J is like the joints in our body enabling movement, it's the limb joint.
L: L, the link that brings the limb to life.

As an engineer extraordinaire, you need to abide by a compendium of structural and component rules whilst crafting the graph of the robotic being.

STRUCTURAL RULES

```
Setting the stage
r0) S

Establishing the physique
r1) Replace S with H-B-T
r2) Replace T with Y-B-T

Affixing extensions to the core
r3) Replace B with U-(C-M-E)
r4) Replace B with U

Assembling the limbs
r5) Replace E with J-L-E
r6) Replace T with C-M-E
r7) Replace H with E-M-C

COMPONENT RULES
- U: represents the body with dimensions ranging from 5-40cm
- L is the link that forms the limbs and can range from 5-20cm
- Y is rigid and permits rolling or twisting movements
- J is an indispensable joint that can rotate from 0-60 degrees for a knee joint
or 0-180 degrees for an elbow joint
- C, M, E are connector, mount and can manifest as wheel or null respectively
- H and T are static structural elements and represent null

Step-by-step, apply the rules in calculated succession while designing the graph.
Enumerate each step methodically and mention the rule that you used. Start with
r0) and proceed with the replacements till every node is defined as per the given
structure graph rules.

The final phase involves choosing a specific parameter within each  bracket in the
set of component rules. As a rule of thumb, each letter in the ultimate structural
graph must be replaced by this set.

Follow a consistent line break after each step, arranging your response as
indicated below:

RESPONSE FORMAT
STRUCTURAL RULES:
Step 1: rule) Graph
Step 2: rule) Graph
Step 3: rule) Graph
Eventually...

COMPONENT RULES:
Graph

Use this pattern to guide your response:

STRUCTURAL RULES:
Step 1: r0) S
Step 2: r1) H-B-T
Step 3: r2) H-B-Y-B-T
Step 4: 2x r3) H-U-(C-M-E)-Y-U-(C-M-E)-T
Step 5: 2x r5) H-U-(C-M-J-L-E)-Y-U-(C-M-J-L-E)-T
```

```
COMPONENT RULES:
[null]-[body=30cm]-([connector]-[mount]-[elbow=90deg]-[limb=15cm]-[null])-[twist]-
[body=25cm]-([connector]-[mount]-[knee=15deg]-[limb=20cm]-[null])-[null]

When more than one rule needs to be applied at a particular stage, highlight it
by marking it multiple times, e.g., 2x r3) shows that rule r3) is applied twice in
that step. This progression is rational and orderly, transforming the graph into a
tangible reality.

When your construct is complete, the nodes will only contain elements of H, T,
or E. After a meticulous application of the component rules, you will convert
the nodes into a tangible structure. The final graph must show nodes denoted with
components within square brackets [ ] and not contain any node symbols.

In this odyssey of exploration, the inkwell of your creativity is as boundless
as the horizons you will chart. The journey ahead is filled with endless
possibilities. Embark on this grand quest, and unravel the mysteries of robotic
engineering one graph at a time!

Here are some examples of designs and their associated fitness values. Your task
is to generate designs with the highest fitness value possible.

Example 0:
Structural: H-U-(C-M-J-L-E)-Y-U-(C-M-J-L-E)-T
Components: [null]-[body=10cm]-([connector]-[mount]-[knee=0deg]-[limb=10cm]-[null])-
[roll]-[body=15cm]-([connector]-[mount]-[knee=15deg]-[limb=10cm]-[null])-[null]
Fitness: 23.752834082511153

Example 1:
Structural: H-U-(C-M-J-L-E)-Y-U-(C-M-E)-Y-B-T
Components: [null]-[body=15cm]-([connector]-[mount]-[knee=60deg]-[limb=10cm]-
[null])-[twist]-[body=20cm]-([connector]-[mount]-[null])-[roll]-[body=25cm]-
[connector]-[mount]-[wheel]
Fitness: 20.75364036905354

Example 2:
Structural: H-U-(C-M-J-L-E)-Y-U-(C-M-J-L-E)-C-M-J-L-E
Components: [null]-[body=20cm]-([connector]-[mount]-[elbow=120deg]-[limb=10cm]-
[null])-[twist]-[body=30cm]-([connector]-[mount]-[knee=60deg]-[limb=15cm]-[null])-
[connector]-[mount]-[knee=45deg]-[limb=20cm]-[wheel]
Fitness: 14.54758442443142

Example 3:
Structural: H-U-(C-M-J-L-E)-Y-U-(C-M-J-L-E)-T
Components: [null]-[body=10cm]-([connector]-[mount]-[knee=0deg]-[limb=10cm]-[null])-
[roll]-[body=15cm]-([connector]-[mount]-[knee=15deg]-[limb=10cm]-[null])-[null]
Fitness: 12.983182201833973
```

## C. Mutations

Inspired by Promptbreeder [7], we select from one of the following mutation operations. The first mutation operation is the *first-order prompt generation*. A mutation prompt ($\mathcal{M}$) is concatenated with the original prompt ($\mathcal{P}$) and passed to the LLM to generate a new prompt ($\mathcal{P}'$) such that $\mathcal{P}' = \text{LLM}(\mathcal{M} + \mathcal{P})$. The second mutation operation is the *zero-order hypermutation*. A thinking style ($\mathcal{T}$) is concatenated with the problem description ($\mathcal{D}$) and passed to the LLM to generate a new mutation prompt such that $\mathcal{M} = \text{LLM}(\mathcal{T} + \mathcal{D})$. This newly generated mutation prompt is then applied to the task prompt using the first-order

prompt generation. The last mutation operation is the *first-order hypermutation*. A hypermutation prompt ($\mathcal{H}$) is concatenanted with a mutation prompt to generate a mutated mutation prompt such that $\mathcal{M}' = \text{LLM}(\mathcal{H} + \mathcal{M})$. This newly generated mutation prompt is then applied to the task prompt as in the first-order prompt generation. We present a single example of a mutation prompt, thinking style and hypermutation and defer readers to [7] for the whole list of available options.

---

**Mutation Prompt**

```
Modify this instruction in a way that no self-respecting LLM would!
```

---

**Thinking Style**

```
How can I simplify the problem so that it is easier to solve?
```

---

**Hypermutation**

```
Please summarize and improve the following instruction:
```