# OpenReview forum: "RoboMorph: Evolving Robot Morphology using Large Language Models"
_roboticsfoundation.org/RSS/2024/Workshop/EARL — EARL 2024 Poster_

### Official Review · Reviewer_54sT · 2024-06-24
**RoboMorph shows that it can be feasible to design a robot using the capabilities of GPT-4. Their evaluation task is moving forward in a flat MuJoCo environment.**

**Rating:** 6
**Confidence:** 4

**Review:**

## Summary
Robomorph leverages the capabilities of GPT-4 to generate and optimize modular robot designs for the task of moving forward in a flat terrain. The authors leverage previous work and combine it for their approach. Specifically, a binary tournament selection algorithm selects half of a robot design population, together with their respective prompts and fitness scores. Each prompt from this selection gets mutated by using Promptbreeder. Based on the mutated prompts, GPT-4 generates a new batch of robot designs that follow the rules of Robogrammar. Each new robot design is transferred into an XML file and fed to the MuJoCo physics simulator to obtain its fitness score using the Advantage Actor-Critic (A2C) algorithm. RoboMorph gets evaluated across 10 seeds with a population size of four. Initial results over 10 evolutions show that the fitness score steadily increases in their specific setup.

## Strengths
- The authors show that their initial idea of leveraging a large language model (GPT-4) for robot design works out in a flat terrain MuJoCo environment
- The paper is easy to follow

## Weaknesses
- The contribution of the authors is not clear. I understand the contribution to be the empirical proof that an LLM can design a robot for the task of moving forward in a flat terrain. This contribution is hidden in proposing an entire robot design framework, consisting of an evolutionary algorithm, Promptbreeder, Robogrammar, and RL-based control. As all of these components are based on previous work and no design choice ablations are provided, the claimed contribution of a robot design framework should be put into question.
    - In my opinion, the authors should focus on the empirical proof that an LLM can design robots for a given task and state that the provided pipeline is the first prototype to achieve their goal.
- The related work section is weak and does not place RoboMorph well into the landscape of related work. The section should be better structured, i.e. the research area that gets summarized in each paragraph should be put in bold at the beginning. Introduced related work should be clearly described and the difference to RoboMorph should be precisely pointed out.
- No ablations of design choices are performed. No baseline algorithms are being introduced and compared.
- It is not clear if RoboMorph is applicable in a different environment setting or with a different task at hand. Therefore, I do not see the contribution as a proof of concept but as a first test of an initial idea.
- No weaknesses of the introduced robot design framework are stated. It is not clear how brittle or robust this initial idea performs and how well it can scale to different experiment settings.

## Improvements
- Figure 1 is not very intuitive for me and could be extended over both columns, including a picture of a robot design population as well as your custom MuJoCo environment. In my opinion, you should already mention Promptbreeder, Robogrammar and A2C in this graphic.
- Your text description of the experiments states “six seeds, over five evolutions and with a population size of four”. Figure 2 provides different information.

I really like the initial idea of validating if a LLM can be used for designing a robot for a given task. However, in my opinion, the current draft of the paper claims too large contributions and should be restructured. I understand that EARL is asking for extended abstracts of 2-4 pages. Therefore, I do not expect an in-depth evaluation in multiple different environments and against multiple different baselines. Still, the following changes are very necessary in my opinion:

- The actual contributions should be narrowed down
- The related work section should better distinguish RoboMorph from prior work
- The experiment section should be extended to at least have a comparison to a single baseline such as Robogrammar. This comparison is crucial in evaluating the initial idea.
- The strengths and weaknesses of the current idea should be stated more precisely such that the reader gets a better impression of its value.

---

### Decision · Program_Chairs · 2024-06-24

Accept (Poster)